# Genetic Diversity in Diospyros Germplasm in the Western Caucasus Based on SSR and ISSR Polymorphism

**DOI:** 10.3390/biology10040341

**Published:** 2021-04-19

**Authors:** Lidia S. Samarina, Valentina I. Malyarovskaya, Stefanie Reim, Natalia G. Koninskaya, Alexandra O. Matskiv, Gregory A. Tsaturyan, Ruslan S. Rakhmangulov, Ruset M. Shkhalakhova, Ekaterina S. Shurkina, Raisa V. Kulyan, Zuhra M. Omarova, Magomed D. Omarov, Alexey V. Ryndin

**Affiliations:** 1Federal Research Centre the Subtropical Scientific Centre of the Russian Academy of Sciences, 354002 Sochi, Russia; malyarovskaya@yandex.ru (V.I.M.); natakoninskaya@mail.ru (N.G.K.); matskiv_a@mail.ru (A.O.M.); grisha.tsaturyan@yandex.com (G.A.T.); rakhmaruslan@yandex.ru (R.S.R.); shhalahova1995@mail.ru (R.M.S.); shurkina-ekaterina@rambler.ru (E.S.S.);supk-kulyan@vniisubtrop.ru (R.V.K.); zuly_om@mail.ru (Z.M.O.); lab-bfbr@vniisubtrop.ru (M.D.O.); ryndin@vniisubtrop.ru (A.V.R.); 2Institute for Breeding Research on Fruit Crops, Julius Kühn-Institut, Federal Research Centre for Cultivated Plants, 01326 Dresden, Germany; stefanie.reim@julius-kuehn.de

**Keywords:** *Diospyros*, persimmon, microsatellite, ISSR-marker, amplified fragment, polymorphism, genotyping, genetic diversity

## Abstract

**Simple Summary:**

Persimmon is an edible fruit consisting of several species in the genus Diospyros. The most widely cultivated and most commercially important species is the Oriental persimmon, *Diospyros kaki*. However, the inter- and intra-specific genetic diversity of the genus Diospyros remains largely unclear and is of great interest both for conservation and breeding purposes. This study describes the genetic diversity and genetic admixture of *Diospyros* germplasm in the Western Caucasus. The information can be used to support conservation measures and the breeding of persimmon.

**Abstract:**

Persimmon germplasm in the Western Caucasus represent one of the most northerly collections. In our study, 51 commercial cultivars of *D. kaki*, 3 accessions of *D. virginiana* and 57 *D. lotus* accessions from six geographically distant populations were investigated using 19 microsatellite and 10 inter simple sequence repeat (ISSR) markers. After STRUCTURE analysis, the single accessions of *Diospyros* were allocated to three genetic clusters. Genetic admixtures in the important genotypes of *D. kaki* were revealed, whereas *D. lotus* accessions showed no admixture with other genetic clusters. The correspondence of genetic data and phenotypical traits was estimated in the *D. kaki* collection. The most frost tolerant genotypes of the collection, such as “Mountain Rogers”, “Nikitskaya Bordovaya”, “Rossiyanka”, “MVG Omarova”, “Meader”, “Costata”, “BBG”, and “Jiro”, showed a high percentage of genetic admixtures and were grouped close to *D. virginiana*. Some of these genotypes are known to be interspecific hybrids with *D. virginiana*. A low level of genetic diversity between the distant *D. lotus* populations was revealed and it can be speculated that *D. lotus* was introduced to the Western Caucasus from a single germplasm source. These results are an important basis for the implementation of conservation measures, developing breeding strategies, and improving breeding efficiency.

## 1. Introduction

The genus persimmon *Diospyros* (*Ebenaceae*) comprises more than 400 evergreen and deciduous tree species that are distributed worldwide [1]. Among these species. *Diospyros kaki* is an important fruit crop and its worldwide production was 5.2 M tonnes with 1.0 Mha of area harvested in 2014, which means five times more production and six times more cultivated area than 30 years ago [2]. *Diospyros lotus* L., also called date plum persimmon, is another important species native to the Balkans, Caucasia to China and Japan [3], and it has been used as rootstock for *D. kaki* as well as for food and medicine [4,5]. The wild species *D. virginiana* (American persimmon) is native to the USA and usually used as timber wood. This species is important as a genetic resource and is usually crossed with *D. kaki* in breeding programs aimed at cold tolerance and early harvesting [6,7].

Germplasm collections outside of the main production countries (for example, in the Western Caucasus) could be a source of genetic diversity to support crop improvement and botanical research. Such border growing regions can be useful for the search of new genetic resources to widen the genetic base for the breeding of highly resistant genotypes, which can be more adaptable to changing environmental conditions. Local genotypes offer a broad range of adaptability to different agro-climatic conditions and therefore are an important resource for breeding purposes. Thus, the valuable genetic resources of the local genotypes should be conserved for their proper utilization. In order to achieve this, it is imperative to assess the genetic variability that exists among local genotypes [5]. Another prerequisite is the disclosure of the genetic background of existing cultivars and wild forms within the germplasm collection, which allows a directed breeding process. Therefore, the accumulation of genetic information for revealing the genetic diversity and genetic integrity of *Diospyros* plants is in urgent need [8,9].

The Western Caucasus is one of the most northerly persimmons growing areas in the world. Persimmon was first introduced in the South-West Caucasus between 1880 and 1890. The set of cultivars from Italy and Japan was introduced and propagated first in Georgia [10]. From Georgia, persimmon spread to nearby experimental stations where breeding efforts had started and about 100 local landraces were derived. Nowadays, three species are grown in these regions: *D. lotus* (2n = 2x = 30), *D. virginiana* (2n = 6x = 90) and *D. kaki* (2n = 6x = 90; 2n = 9x = 135). *D. lotus* and *D. virginiana* are the most cold-tolerant species in the genus *Diospyros* and are important for the development of new drought or cold-tolerant varieties [11]. The standard assortment for this region included famous cultivars such as “Hachiya”, “Hiyakume”, and “Zenjimaru”. The crop commercially spread across nearby Caucasus in the following regions: Krasnodar region, Dagestan Republic, Georgia Republic, Abkhazia Republic, South-West Turkmenistan, Tajikistan, Uzbekistan, Azerbaijan, and Crimea. Dozens of locally adapted cultivars were developed over the 100-year history of the conventional persimmon breeding in the Western Caucasus. Now, the local collection includes a set of frost tolerant hybrids and cultivars of persimmon, with some surviving extreme temperatures up to −20 °C with good yields [8]. However, the intraspecific genetic diversity and phylogenetic relationships of these germplasm collections still have not been evaluated.

In *Diospyros*, molecular markers have been developed in recent years for the efficient characterization and further utilization of germplasm resources. Phylogenetic relationships of *Diospyros* species are mainly studied based on random amplification of polymorphic DNA (RAPD) [11], simple sequence repeats (SSRs) [12,13,14,15,16], internal transcribed spacer (ITS) [17], sequence-related amplified polymorphism (SRAP) [14,18], inter-retrotransposon-amplified polymorphism (IRAP) [19], start codon-targeted (SCoT) [19,20,21] or organelle sequences (orgDNA) [22]. 

Among the different marker types, co-dominant nuclear microsatellite markers (SSRs) have desirable advantages for assessing the genetic features of species at individual and population levels, such as locus specificity, high reproducibility, high polymorphism, and technical simplicity [23,24]. However, the genetic diversity can be underestimated due to homoplasy, as reported elsewhere [25,26]. The combination of several marker types can be helpful for the better understanding of phylogenetic relationships in the collections.

Inter simple sequence repeats (ISSRs) are another efficient marker type that is multilocus, reproducible and highly polymorphic for genetic diversity studies [27]. ISSRs are believed to be present mostly in the non-coding regions of chromosomes and specific stretches of DNA sequences which are not active [28]. Although ISSRs are dominant molecular markers, several informative sequences could be generated using specific ISSR primers, with larger anchorage regions for SSR loci. High repeatability could be observed for ISSR primers with sufficient specificity [29,30]. The high occurrence of ISSRs between normal coding genes and their presence in certain chromosomes as satellite bodies make ISSRs unique and advantageous to be used for DNA fingerprinting. However, no efficient ISSRs have been reported for *Diospyros* genotyping to date. The current study presents for the first time the successful usage of ISSR markers for the genetic analysis of *Diospyros*. Thus, ISSR markers can be useful to evaluate genetic diversity within cultivated and naturalized populations of *Diospyros* germplasm.

The aim of our work was to evaluate the genetic diversity, genetic structure and phylogenetic relationships in *Diospyros* collection in the Western Caucasus using SSR and ISSR markers. The obtained results are an important basis for the implementation of conservation measures, developing breeding strategies, and improving the breeding efficiency.

## 2. Materials and Methods

### 2.1. Plant Material and DNA Extraction

The persimmon core collection of the SSC RAS (Sochi, Russia) containing 51 accessions of *D. kaki*, 3 accessions of *D. virginiana*, and 57 accessions of *D. lotus* from 7 geographically distant populations was investigated in this study (Figure 1; Appendix A). Phenotypical evaluation of the core collection was made in the period from 1990–2020 in field conditions (Figure 1—red flags). Five trees of each cultivar were evaluated during the years, and summary data are presented in the table as binary data indicating the presence (1) or absence (0) of each trait (Appendix A). Young and healthy leaves of each accession were collected in 2 mL tubes and dried using silica gel. The leaf material was stored at 4 °C until DNA isolation. The dried leaf material was ground and DNA extraction was performed using the CTAB protocol [31]. DNA quality was checked by agarose-gel electrophoresis and spectrophotometrically and all samples were diluted to 20 ng/µL and stored at −20 °C. 

### 2.2. Genetic Analysis

For the SSR analysis, 19 nuclear microsatellite (SSR) primer pairs, developed for *Diospyros* [15,16], were tested (Appendix A) Additionally, the transferability of 10 ISSR primers developed for the other species [32,33] was evaluated to study polymorphisms in ISSR genome regions in *Diospyros* (Appendix A). 

PCR for SSR analysis was performed in the 20 μL PCR reaction mixture volume containing 10 μL 2 × HS-TaqPCR reaction buffer (Biolabmix, Russia, biolabmix.ru ), 0.2 μL of each primer (10 µM), 1 μL of DNA (20 ng µL^−1^) and nuclease free water for PCR. Two-step amplification program was used: primary denaturation 5 min at 95 °C, annealing 40 cycles of 15 sec at 50–60 °C and the final elongation at 72 °C for 7 min. The separation of SSR-fragments was performed on a QIAxcel Advanced system (Qiagen, Montgomery County, MD, USA) with the High Resolution kit according to manufacturer instructions.

PCR for the ISSR amplification was performed in the 20 μL PCR reaction mixture volume consisting of 10 μL 2 × HS-TaqPCR reaction buffer including Hot Start Taq-Polymerase (Biolabmix, NO, Russia), 0.3 μL of primer (10 µM), 1 μL of DNA (20 ng µL^−1^) and DEPC-treated water. Amplification was carried out in the MiniAmp thermal cycler (Thermo Fisher Scientific, USA) with the following program: primary denaturation 5 min at 95 °C, annealing 40 cycles of 20 sec at 53 °C with elongations at 72 °C for 1 min 45 sec and the final elongation at 72 °C for 7 min. The separation of ISSR fragments was performed on a 2% agarose gel for 2.5 h at 90 V in 1 × Tris-Acetate buffer. The data were recorded as 1/0 matrix for the presence and absence of amplified fragments, respectively.

### 2.3. Statistical Analysis

Genetic diversity parameters were calculated for each SSR and ISSR locus using the software program GeneAlex ver. 6.5: mean number of alleles by locus (*Na*) and effective number of alleles (*Ne*), the observed heterozygosity (*Ho*), Shannon’s Information Index (I) and for each ISSR the diversity (*h*). Furthermore, the genetic parameters were calculated separately for each genetic cluster. 

The analysis function “Matches” in GeneAlex ver. 6.5 [34,35] was used to identify genotypes with identical allelic patterns within the SSR and ISSR datasets. Subsequently, the model-based clustering method was applied for all genotypes to verify the assignment using the software STRUCTURE ver. 2.3.4. [36]. The run parameters were 50,000 burn-in periods and 50,000 Markov Chain Monte Carlo repetitions using the admixture model. The software program STRUCTURE HARVESTER [37] was used for detecting the most likely value for K based on Evanno’s ΔK method [38]. To examine the genetic structure of the collection, a principal coordinate analysis (PCoA) was performed based on the distance matrix data set in GeneAlex ver. 6.5 with 1000 random permutations. An analysis of molecular variance (AMOVA) was performed using SSR and ISSR marker data. The software DARWIN 6.0 [39] was also used for Neighbor Joining analysis of phenotypical distances in the core persimmon collection.

## 3. Results

### 3.1. Efficiency and Resolving Power of SSR and ISSR Markers 

Eight out of 19 SSR markers showed clear polymorphisms within the *Diospyros* germplasm. These markers were selected for the further analysis of the whole collection (ssrdk01, ssrdk03, ssrdk06, ssrdk09, ssrdk10, ssrdk14, ssrdk26, ssrdk30). 

In the *D. kaki* core collection, an average number of different alleles of *Na* = 8.9 per locus was identified in eight polymorphic loci, ranging from *Na* = 4 for ssrdk1 to *Na* = 16 for ssrdk26 (Table 1). The average effective number of alleles was *Ne* = 4.6. The highest number of effective alleles was calculated for ssrdk26 with *Ne* = 8.5 and the lowest number of effective alleles with *Ne* = 2.3 for ssrdk06. The mean observed heterozygosity was *Ho* = 0.6, with the lowest value for ssrdk06 (*Ho* = 0.3) and the highest value for ssrdk09 (*Ho* = 0.9).

Among *D. lotus* accessions, only three of 19 SSR markers detected clear polymorphism (ssrdk14, ssrdk26, ssrdk30). An average number of different alleles of *Na* = 2.2 per locus was identified in eight polymorphic loci, ranging from *Na* = 1 for ssrdk01, 03, 06, 09, 10 to *Na* = 5 for ssrdk14 (Table 1). The highest number of effective alleles was calculated for ssrdk14 with *Ne* = 2.4. The mean observed heterozygosity was *Ho* = 0.2, with the highest value for ssrdk14 (*Ho* = 0.8).

Of 10 ISSR primers, five ISSRs showed reproducible results with clear polymorphisms and resolution within the *D. kaki* and *D. lotus* genotypes (ISSR13, ISSR15, ISSR814.1; ISSR815, ISSR880) (Appendix A). Other ISSRs showed a low amplification quality and were removed from the analysis. A total of 72 bands were detected with five ISSRs resulting in 94.44% of polymorphic fragments for *D. kaki* and 33.33% of polymorphic fragments for *D. lotus*. The number of bands among the five ISSR markers ranged from *Na* = 8 to 17 (Table 2). The average effective number of alleles was *Ne* = 1.5 for *D. kaki* and *Ne* = 1.1 for *D. lotus*. In the *D. kaki* collection, the mean diversity was *h* = 0.3, with the lowest value for ISSR15 (*h* = 0.2) and the highest value for ISSR814.1 (*h* = 0.4). In the *D. lotus* collection, the mean diversity was *h* = 0.1, with the lowest value for ISSR15 (*h* = 0.0) and the highest value for ISSR13 (*h* = 0.2).

Identical DNA fingerprints were not observed in the *D. kaki* collection, but they were in the *D. lotus* collection. In total, 37 and 33 *D. lotus* accessions with identical SSR- and ISSR-fingerprint patterns were identified, respectively, within populations and among them (Table 2). Generally, ISSR markers revealed less identical genotypes in three of seven locations (Sochi, Gagra, and Gulripsh). No difference in the determination of identical genotypes between SSR and ISSR markers was observed in two locations (Kalinovoe lake, Piket). Interestingly, a higher number of identical fingerprints was observed by ISSR compared to SSR in two locations of *D. lotus* (Shkhafit and Sukhum) (Table 3). 

In addition, some misclassifications were observed in the core collections of *D. kaki* based on SSR and ISSR data. Five *D. kaki* genotypes (“XX Century”, “Jiro”, “Seedless”, “Gosho”, “Geyli1”) that were mostly obtained from the backup core collection (Figure 1) and previously assigned to one and the same cultivar showed different genetic patterns based on both SSR and ISSR markers. Additionally, considering phenotypical traits, these accessions were identified as misclassified cultivars and were excluded from the further analysis. Accessions with a clear assignment of the cultivar name were left in the dataset for further analysis. 

### 3.2. Genetic Diversity and Population Structure of Diospyros Germplasm

Following STRUCTURE analysis, 80 accessions were grouped into three genetic clusters (K = 3) (Figure 2). Cluster 1 contained only *D. lotus* accessions, whereas *D. kaki* accessions were grouped into two clusters (cluster 2 and 3). The 26 *D. kaki* cultivars in cluster 2 showed no genetic admixture with *D. lotus* and *D. kaki* accessions of cluster 3, interestingly. Similarly, cluster 3 included 14 *D. kaki* cultivars that appeared to be genetically different from the *D. kaki* cultivars of cluster 2. However, cluster 3 also included nine cultivars with genetic admixtures of 5–50% of cluster 2 and 1. Particularly, the cultivars “Nikitskaya Bordovaya”, “Meader”, “Fuyu” and “Nitari” showed remarkable genetic admixture of cluster 2. Due to the small accession number of *D. virginiana*, no separate genetic cluster could be found for *D. virginiana*. Instead, the two *D. virginiana* accessions showed 80% of alleles that belong to the *D. kaki* cluster 2 and 20% of the allele frequency was similar to *D. lotus.*

Based on the SSR data, cluster 2 showed the highest observed heterozygosity (*Ho* = 0.62) (Table 4). This was also confirmed by ISSR data that showed the highest polymorphism level *P* = 90.28% in cluster 2. A high number of rare and private alleles was revealed in cluster 3 and 2 (data not shown), indicating a high rate of genetic difference. In both *D. kaki* clusters (2 and 3), the mean polymorphism was 100% based on the SSR data and about 85 and 90% based on ISSR markers, respectively. In contrast, the *D. lotus* accessions (cluster 1) showed low heterozygosity and low genetic diversity based on both SSR and ISSR data. The mean polymorphism of the applied markers was 38% (SSR) and 33% (ISSR) among *D. lotus* accessions. 

### 3.3. Phylogenetic Relationships and Connection with Phenotype in Diospyros Collection

In general, PCoA supported the results of the STRUCTURE analysis and showed a comparable relationship for the three clusters observed using the Bayesian iterative algorithm (Figure 3). Cluster 1 combined all *D. lotus* accessions, which were grouped in their own cluster after PCoA analysis (Figure 3, green circle). However, some *D. lotus* accessions (Dl10, Dl4, Dl3, Dl2) showed a slight separation from the other *D. lotus* accessions, which confirmed the admixtures revealed by STRUCTURE analysis. 

The PCoA-cluster 2 (Figure 3, orange circle right above) combined 27 genotypes including mostly cultivars of Japanese origin, as well as all Russian and Ukrainian cultivars. Most round-square-shaped cultivars joined this group (“Mountain Rogers”, “Khostinskii”, “MVG Omarova”, “Hybrid №39”). The cultivars “Khostinskii”, “MVG Omarova”, and “Hybrid №39” were descendants of the local cultivar “Jiro” (Figure 4). In addition, most dwarf genotypes joined this cluster (“Tanenashi”, “Sayo”, “Tran Takaki”, “Takura”). However, no pollinator, no pollinator constant non astringent (PCNA) genotype, and no large-fruited genotype occurred in this cluster (Figure 3, Appendix A). 

The PCoA-cluster 3 combined 12 genotypes of *D. kaki*, including three main pollen donor cultivars “Zenjimaru”, “Pollinator#8” and “Geyli” (Figure 3, orange circle, right below). No PCNA-genotypes joined this cluster, but most of them were pollinator variable astringent and non-astringent. In addition, all large-fruited genotypes are combined in this cluster (“Tamopan”, “Gosho”, “Hachiya”). Moreover, most of the genotypes in this group were tall or medium tree size, either early harvesting (“Seedless”, “Zenjimaru”, “Tsurunoko”, “Hachiya”, “Geyli”) or late harvesting (“Costata”, “Tamopan”, “Gosho”, “Pollinator#8”) with round or flat-round fruit shape (Figure 3, Appendix A). 

Some accessions (Figure 3, blue symbols), previously identified by STRUCTURE as belonging to cluster 3 (*D. kaki*), showed a distinct group by PCoA and were placed between PCoA-cluster 2 and 3, which included *D. kaki* accessions. On the basis of the PCoA, these three accessions of *D. virginiana* seem to be more genetically distant to *D. lotus* as suggested based on STRUCTURE output. The five accessions of *D. kaki* (“XX Century”, “Jiro”, “Fuyu”, “Nitari”, “Saburousa”) were also not grouped in PCoA-cluster 2 or 3 but between them, indicating a genetic mixture of both clusters. Interestingly, this inter-cluster combined all constant genotypes: PCNA genotypes (“Jiro”, “XX Century” and “Fuyu”) and pollinator constant astringent genotypes (“Saburousa”, “Nitari”, “Meader”) that were completely missing in both *D. kaki* PCoA-clusters. In addition, all genotypes of this inter-cluster are early harvesting and most of them are flat-round fruit shaped (Appendix A). In addition, the most frost tolerant genotypes of the collection such as “Mountain Rogers”, “Nikitskaya Bordovaya”, “Rossiyanka”, “MVG Omarova”, “Meader”, “Costata”, “BBG”, and “Jiro” were grouped closer to *D. virginiana* and some of them are known to be interspecific hybrids with this species.

To determine any relationship between marker and phenotype data, a phylogenetic analysis using DARWIN was performed based on the most important four fruit traits: harvesting period, fruit size, astringency and fruit shape (Figure 4, Appendix A). Based on these traits, the core data set of *D. kaki* accessions was separated into three distant clusters. The phenological traits of many genotypes were clustered according to the genetic clusters obtained based on SSR and ISSR data. For example, “XX Century” and “Fuyu”, with common phenotypical traits such as PCNA, medium fruit size, flat-round fruit shape, and early harvesting, were in one genetic cluster; “Takura”, “Tenjingosho” and “Aizumishirazu” with joint fruit traits such as pollinator constant astringent fruit type, medium fruit mass and long-heart fruit shape were grouped in another genetic cluster. On the other hand, the tree-position of several cultivars did not correlate with the genetic clusters (red colored cultivars, Figure 4). Thus, it can be assumed that not all phenotypical traits follow a grouping according to the genetic cluster. Therefore, the categorization of germplasm accession should be based on phenotypic and genetic traits. 

## 4. Discussion

### 4.1. Efficiency and Resolving Power of SSR and ISSR Markers

Microsatellites selected for our study were developed previously for *Diospyros kaki* and were used by several researchers for genetic diversity studies [12,13,14,15,16]. These microsatellites were derived from the cultivar “Rojo Brillante” [15]. Although multibanding pattern was reported for some of these SSRs [15], we did not observe more than two bands in each individual accession. The numbers of detected alleles for these SSRs that were observed in the different studies differed depending on the size of the dataset investigated. For example, ssrdk01 showed six alleles with *Ho* = 0.73 when the dataset included 12 cultivars [15]. Nevertheless, this marker revealed 12 alleles when the dataset was increased to 71 cultivars [16]. 

Due to the nature of hexa- and nano-ploidy in the *D. kaki* genome [40], the expected heterozygosity (*He*) was not calculated. Our results showed a mean observed heterozygosity (*Ho*) of 0.6, which is highly consistent with the study of Soriano et al. [15] that detected an observed heterozygosity for selected SSRs in the range of 0.27–0.83 [15]. The level of polymorphic alleles in the *D. kaki* collection was also high, with *P* = 100% based on eight SSRs used, and comparable to other studies on *Diospyros* [6,14,16]. However, many markers were not efficient and SSRs are limited for *Diospyros* genotyping [19,41]. Therefore, it is still necessary to develop more efficient SSR markers for *D. kaki* and other *Diospyros* species that can be used for genotyping and evolutionary studies in artificial or natural populations. In addition, the genetic differences revealed by SSR markers can be underestimated because of homoplasy that was observed in some plant species [25,26], hence, the combination of different marker types can be useful for genetic diversity studies.

A higher level of diversity was observed using selected multilocus ISSR markers. Our results showed that efficient ISSRs are placed between the following genome regions in *Diospyros*: (CT)_8_G, (GGAGA)_3_, (AC)_8_C, (TC)_8_C, (CT)_8_TG. In contrast, inefficient ISSRs for *Diospyros* were placed between the following regions: (CT)_8_T, (TAT)_5_, (CTTCA)_3_, (TG)_8_G. On average, 15 bands per ISSR primer in each individual accession were obtained in our *Diospyros* collection with an allele size range of 200–2500 bp using the mentioned PCR conditions. Another study showed that primers with (AG), (GA), (CT), (TC), (AC), (CA) repeats are more polymorphic than primers with other di-, tri- or tetra-nucleotide repeats [27]. Tri and tetra-nucleotides are less frequent, and their use in ISSRs is lesser than the di-nucleotides. The (AG) and (GA) based primers have been shown to amplify clear bands in several crops, such as rice, citrus, chickpea, etc., whereas primers based on (AC) di-nucleotide repeats were found to be more useful in wheat and potato [27]. However, based on our results, we can confirm the successful usage of tri and tetra-nucleotide primers as well as the effective usage of primers with (AC) di-nucleotide repeats in *Diospyros*. 

### 4.2. Genetic Diversity and Population Structure of Diospyros Germplasm

Bayesian analysis showed homogeneity in *D. lotus* accessions, although all populations were geographically distant. Furthermore, the genetic diversity was low. These results are not consistent with previous data obtained with RAPD markers [5]. In addition, other authors used SCoT markers and indicated higher genetic variation among studied *D. lotus* accessions [21]. The low genetic diversity can be attributed to the occurrence of a high number of genetically identical accessions in the populations investigated. It is plausible that *D. lotus* was introduced to the Western Caucasus from a single germplasm source, hence the homogeneity, low genetic diversity and the formation of a private genetic cluster without significant admixtures. 

In contrast, we observed the clear separation of *D. kaki* germplasm into two different clusters using SSR and ISSR markers. Another study on *D. kaki* germplasm using SSR, IRAP and SCoT markers separated 228 accessions into five genetic clusters mostly based on geographical origin [6,19]. In comparison, our results revealed lower cluster numbers in our *D. kaki* collection. However, remarkably fewer *D. kaki* accessions were analyzed in our study, which probably had an influence on the genetic structure. Cluster 2 was characterized by greater homogeneity with less genetic admixtures compared with cluster 3. Cluster 2 mostly combined the cultivars with Japanese origin or locally derived cultivars. In contrast, cluster 3 consist mostly of cultivars that originated from Japan, China or USA.

*D. virginiana* accessions have been used actively in breeding programs in Russia and Ukraine to develop high cold-tolerant and disease-tolerant cultivars. *D. virginiana* accessions “Saburouza” and “Meader” joined cluster 3 and showed about 10% of *D. lotus* admixture. However, the genetic relationship with *D. lotus* appears implausible and could not be confirmed by the PCoA analysis. It is probable that these accessions were only assigned to the *D. lotus* cluster because of the small number of *D. virginiana* accessions in the study. “Meader” is known as *D. virginiana* species, but “Saburouza” is related to *D. kaki* species [42]. Based on ISSR data, both cultivars have a similar genetic pattern that is related to *D. virginiana*. 

Several genotypes with remarkable genetic admixtures of cluster 3 and cluster 2 were detected, namely, “Nikitskaya Bordovaya”, “Fuyu” and “Nitari”. “Nikitskaya Bordovaya” was believed to be developed by interspecific hybridization between *D. virginiana* and *D. kaki* [11]. However, we have not revealed the same allele distribution as in the other *D. virginiana* accessions, which is probably also attributed to the small number of *D. virginiana* accessions in our study. 

Surprisingly, “Rossiyanka” was grouped to cluster 2, although it is known to be a interspecific hybrid of *D. virginiana* and *D. kaki* [11]. However, since our results showed no genetic admixtures, we speculate that it belongs to *D. kaki*. The same is true for “MVG Omarova” (cluster 2), which was assumed to be a hybrid of “Jiro” and “Meader” but showed only low interspecific admixture. 

“Fuyu” is an important world cultivar and important parental genotype for persimmon breeding in the Western Caucasus [7]. This can be the reason for the high ratio of alleles from both cluster 3 and cluster 2 observed in this genotype by STRUCTURE analysis. “Fuyu” is a cultivar of Japanese origin with a high pollen fertility level and is often used in Russian breeding programs aimed at early harvesting, non-astringency and the medium tree size [43]. Similar admixture profiles were observed in “Nitari” and “Nikitskaya Bordovaya”, although these genotypes have not been used as parents. These cultivars are probably closely related to “Fuyu” and therefore show similar admixture profiles. Further investigation is necessary for a definite conclusion to be made. In addition, “XX Century” and “Jiro” belonged to cluster 3, but have some admixture from cluster 2. “Jiro” was introduced from Japan in 1935 and it has male sterility, but it can be used as a maternal genotype that allows the breeding of new large-fruited PCNA varieties. 

### 4.3. Phylogenetic Relationships and Connection with Phenotype in Diospyros Collection

The main problem of persimmon breeding is the low pollen fertility, hence most cultivars can used only as a mother parent in breeding purposes. In the Western Caucasus, only the cultivars “Zenjimaru”, “Geyli”, “Fuyu” and “Pollinator#8” were used as parental genotypes in breeding programs because they have sufficient fertile pollen. As shown in Figure 3, three of these cultivars except “Fuyu” were grouped together in PCoA-cluster 3. However, “Fuyu” is placed between PCoA-cluster 2 and 3 because it is the paternal genotype for most of the cultivars. 

Most accessions in cluster 2 are phenotypically similar and show a dwarf tree phenotype, not big flat fruits with variable astringency and early harvesting period. Interestingly, the cultivars “Fuyu”, “Nitari”, “Nikitskaya”, “XX Century” and “Jiro” have similar phenotypical traits and also showed a close relationship based on ISSR analysis. This result leads us to speculate that these ISSR markers can be useful to find associations with the important traits in persimmon genotypes. In addition, the most frost-tolerant genotypes of the collection, such as “Mountain Rogers”, “Nikitskaya Bordovaya”, “Rossiyanka”, “MVG Omarova”, “Meader”, “Costata”, “BBG”, and “Jiro”, were grouped closer to *D. virginiana* and some of them are known to be its interspecific hybrids, which was confirmed by ISSR analysis. However, some other frost-tolerant genotypes such as “Pollinator#8” and “Hyakume” were jointly grouped in the distant cluster.

Nevertheless, our results showed no clear clustering by any certain trait (for example, cold tolerance or fruit shape). However, several phenotypical traits considered jointly allowed us to reveal the correlation between phenotypical data and the genetic clusters obtained. For example, the trait combination non astrigent+early ripening+flat-round fruit shape was common for most cultivars clustered in between cluster 2 and 3. Several cultivars (“Tran Takaki”, “Pollinator#8”, “Hiyakume”, “Hachiya”, “Seedless”) were not clustered on the phenotypical tree in accordance with molecular data. Thus, it can be concluded that not all phenotypical traits follow a grouping according to the genetic cluster. These results indicate that a categorization of germplasm accessions should be based on phenotypic and genetic traits. 

## 5. Conclusions

Our results showed the efficiency of the SSR and ISSR markers for inter- and intraspecific genetic diversity studies in *Diospyros* spp. Low genetic diversity and the formation of a private genetic cluster without significant admixtures were observed in the distant *D. lotus* populations. Therefore, new efforts are necessary to introduce this germplasm from the other world sources to prevent genetic erosion. The most frost-tolerant genotypes showed a high percentage of genetic admixtures and were grouped close to *D. virginiana*. Some of them are known to be interspecific hybrids with this species, which was confirmed by our study. The correspondence between the fruit traits and genetic data was observed, particularly with fruit astringency. The results will be useful for further studies to find associations between the markers and phenotypical traits in Diospyros germplasm.

## Figures and Tables

**Figure 1 biology-10-00341-f001:**
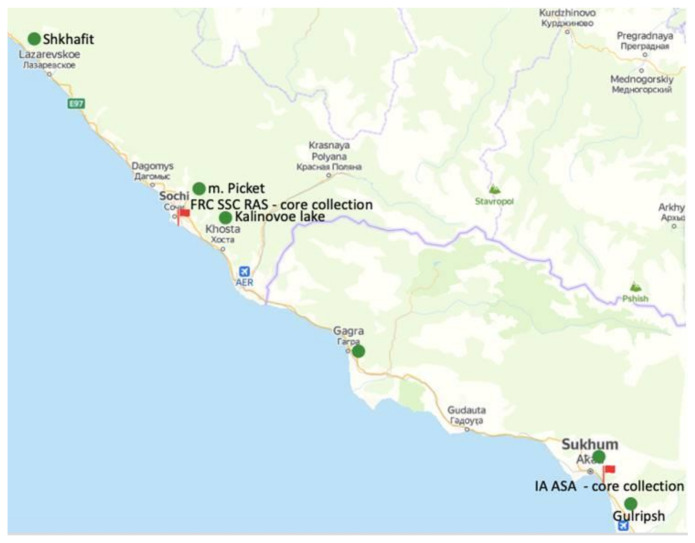
Geographical origin of *D. lotus* populations (green circles) and location of core collection of *D. kaki* accessions (red flags) included in the study.

**Figure 2 biology-10-00341-f002:**
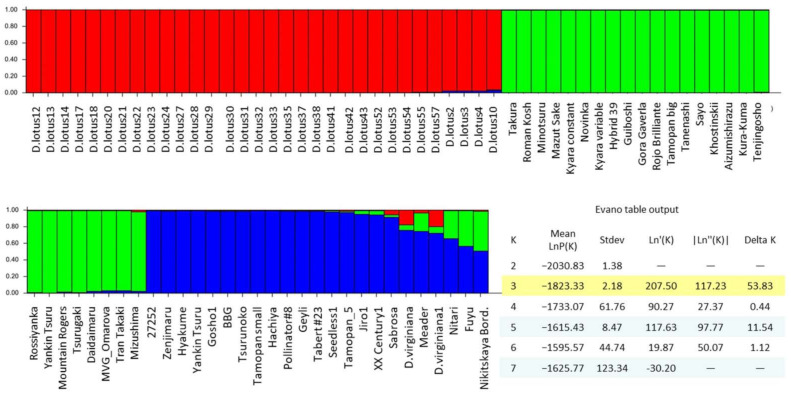
Population genetic structure among 80 *Diospyros* accessions based on 72 ISSR loci. Each genotype is represented by a vertical bar partitioned into *K* = 3. Each color represents the estimated membership fraction of the three genetic clusters (cluster 1 = red, cluster 2 = green, cluster 3 = blue).

**Figure 3 biology-10-00341-f003:**
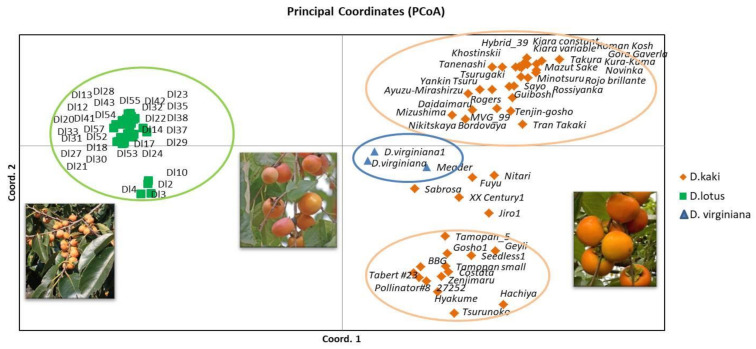
Principal coordinate analysis (PCoA) of pairwise distances between accessions calculated on ISSR markers.

**Figure 4 biology-10-00341-f004:**
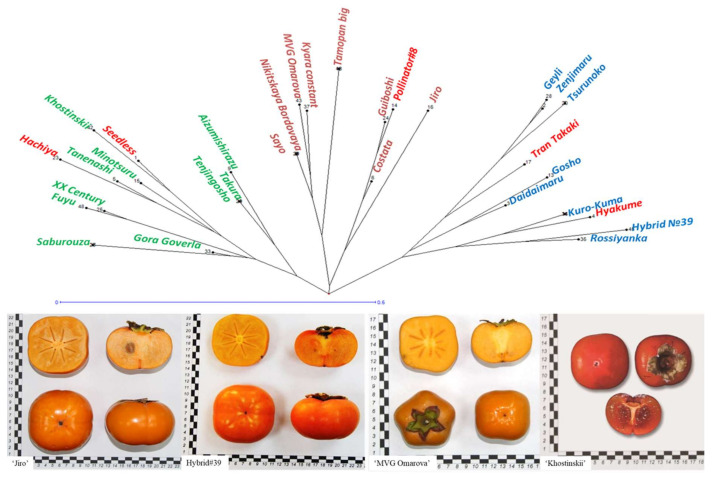
Phenotypical variability and Neighbor-Joining clustering of core *D. kaki* collection based on the following traits: fruit astringency, fruit size, fruit shape and harvesting period. Red cultivars—placement is not in accordance with molecular data.

**Table 1 biology-10-00341-t001:** Genetic diversity parameters for 8 SSR loci used.

Species	SSR	N	*N_a_*	*N_e_*	*I*	*H_o_*
*D. kaki*	ssrdk14	32	11.000	5.611	1.977	0.438
	ssrdk26	33	16.000	8.475	2.433	0.606
	ssrdk30	30	7.000	3.666	1.564	0.700
	ssrdk01	36	4.000	2.883	1.142	0.417
	ssrdk03	33	6.000	2.339	1.096	0.667
	ssrdk06	33	6.000	2.276	1.138	0.333
	ssrdk09	31	10.000	4.319	1.721	0.871
	ssrdk10	36	11.000	6.968	2.110	0.694
MEAN ± Standard error		33.0 ± 0.8	8.9 ± 1.4	4.6 ± 0.8	1.6 ± 0.2	0.6 ± 0.1
*D. lotus*	ssrdk14	31	5.000	2.427	1.041	0.839
	ssrdk26	30	3.000	1.753	0.675	0.600
	ssrdk30	28	3.000	1.332	0.464	0.071
	ssrdk01	32	1.000	1.000	0.000	0.000
	ssrdk03	32	1.000	1.000	0.000	0.000
	ssrdk06	32	1.000	1.000	0.000	0.000
	ssrdk09	32	1.000	1.000	0.000	0.000
	ssrdk10	32	1.000	1.000	0.000	0.000
MEAN ± Standard error		31.1 ± 0.5	2.0 ± 0.5	1.3 ± 0.2	0.3 ± 0.1	0.2 ± 0.1

*N*: number of analyzed individuals; *N_a_*: number of different alleles; *N_e_*: number of effective alleles (=1/(∑*p_i_*^2^)); *p_i_*: relative frequency of the *i*th allele; *I* = Shannon’s information index = −1 * Sum (pi * Ln (pi)); *H_o_*: observed heterozygosity (=number of heterozygotes/N).

**Table 2 biology-10-00341-t002:** Genetic diversity parameters for the selected ISSR markers.

Species	Marker	No bands	Av. Band Freq.	N	*N_a_*	*N_e_*	*H*
*D. kaki*	ISSR815	16	0.390	56.000	1.750	1.532	0.307
	ISSR880	16	0.259	56.000	2.000	1.545	0.323
	ISSR13	8	0.500	56.000	2.000	1.600	0.360
	ISSR814.1	17	0.370	56.000	2.000	1.705	0.402
	ISSR15	15	0.368	56.000	1.733	1.241	0.169
MEAN ± Standard error		14.6 ± 3.0	0.1 ± 0.0	56.0 ± 0.0	1.9 ± 0.1	1.5 ± 0.0	0.3 ± 0.0
*D. lotus*	ISSR815	16	0.408	32.000	0.938	1.104	0.073
	ISSR880	16	0.502	32.000	1.313	1.130	0.094
	ISSR13	8	0.383	32.000	1.625	1.213	0.160
	ISSR814.1	17	0.309	32.000	0.824	1.163	0.096
	ISSR15	15	0.867	32.000	0.867	1.000	0.000
MEAN ± Standard error		14.4. ± 3.0	0.5 ± 0.1	32.0 ± 0.0	1.1 ± 0.1	1.1 ± 0.0	0.1 ± 0.0

*N*: number of analyzed individuals; *N_a_*: number of different alleles; *N_e_*: number of effective alleles (=1/(∑*p_i_*^2^)); *p_i_*: relative frequency of the *i*th allele; *h* = Diversity = 1 − Sum pi^2^.

**Table 3 biology-10-00341-t003:** Identical allelic pattern of *D. lotus* accessions based on ISSR and SSR analysis.

Group	Population Location	Number of Accessions in the Population	Number of Identical Accessions Based on SSR Data	Number of Identical Accessions Based on ISSR Data
1	Shkhafit	6	4	6
2	Piket	4	3	3
3	Sochi center	12	6	5
4	Kalinovoe lake	1	0	0
5	Gagra	19	12	7
6	Sukhum	10	4	8
7	Gulripsh	5	8	4
	TOTAL	57	37	33

**Table 4 biology-10-00341-t004:** Genetic diversity parameters for each genetic cluster based on SSR and ISSR markers.

		SSR				ISSR			
Cluster	*N*	*N_a_*	*N_e_*	*H_o_*	*% P*	*N_a_*	*N_e_*	*h*	*% P*
1 (*D. lotus*)	32	2.00	1.31	0.20	37.50	1.06	1.11	0.08	33.33
2 (*D. kaki*)	26	7.13	4.13	0.62	100.00	1.81	1.44	0.27	90.28
3 (*D. kaki* and hybrids)	23	5.13	3.44	0.52	100.00	1.71	1.51	0.29	84.72
mean		4.75	2.96	0.44	79.17	1.52	1.35	0.21	69.44

*N*: number of analyzed individuals; *N_a_*: mean number of different alleles in the cluster; *N_e_*: mean number of effective alleles in the cluster (=1/(∑*p_i_*^2^)); *H_o_*: observed heterozygosity (=number of heterozygotes/N); *h* = Diversity (1 − Sum pi^2^); *P* = Percentage of Polymorphic Loci.

## Data Availability

Data are contained within the article or Appendix A.

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
