# Peer review of "Genetic Diversity in Diospyros Germplasm in the Western Caucasus Based on SSR and ISSR Polymorphism"

_biology, 2021, doi:10.3390/biology10040341_

Round 1

Reviewer 1 Report

The authors have addressed all the issues, and the MS has significantly been improved. I have no more comments. Thus, I am endorsing the resubmission for publication in Biology. Please address the below-mentioned comment before official acceptance.

  • Page 11, please move the conclusion section at the end of the methodology with proper heading, i.e., 5. Conclusion.

Congratulations!

Author Response

  • Dear Reviewer, we appreciate your help and your support during the submission process. We wish you good luck and success in your research.

Reviewer 2 Report

I like the current version of the manuscript. I am impressed by the changes made within the text structure. Now it looks more professional. Nevertheless, the number of markers is really small and I don't recommend hypothesising based on this data. Please, read once again your text and try to eliminate speculation, especially if your data shows some controversial results.

I have some minor concerns:

Line 27: closer - close, or closer to D. v. than...

Line 61: 18801-1890.

Line 132: in - delete

Line 358: chowed - showed

Line 380: different - others

Line 382: closer - close

Table 5 and 6 should be supplemental.

Author Response

  • Dear Reviewer, thank you very much for careful revision of our manuscript, thanks for your suggestions and comments. We read our text again and tried to correct our article accordingly.

I have some minor concerns:

Line 27: closer - close, or closer to D. v. than...

  • Revised

Line 61: 18801-1890.

  • Fixed

Line 132: in – delete

  • I have not found this

Line 358: chowed – showed

  • Fixed

Line 380: different – others

  • Revised

Line 382: closer – close

  • Fixed

Table 5 and 6 should be supplemental.

  • Done

This manuscript is a resubmission of an earlier submission. The following is a list of the peer review reports and author responses from that submission.

Round 1

Reviewer 1 Report

In the current manuscript, the authors examined the genetic diversity and phylogenetic relationships in Diospyros germplasm using SSR and ISSR markers. This study was nicely planned and executed. Apart from the genetic diversity analysis, it would have been nice addition considering some traits/features (such as frost tolerance) with the studied plants' genetic makeup. However, the current version needs improvement.

  • Line 14, please mention the full name on the first appearance in the abstract (D. kaki).
  • Line 20, define the abbreviations (SSR and ISSR) on the first appearance in the abstract and the text.
  • Line 37 and 59, please cite “FAOSTAT, 2017” according to other references and add it to the reference list.
  • In fig S1, marker ISSR880, please label the gel at the top of the image as other markers. Further, the gel image of ISSR815 is missing. Please explain the meaning of numbers, M, and C that are presented on the image.
  • Line 158, 185, 263, italic > D. lotus/D. Kaki. Please check the entire text for this error, if any.
  • Line 180-181, add a space between D. lotus.
  • Line 279, change Soriano et al. (2006) to Soriano et al. [16].
  • Line 386, fix superscript in 4o C.
  • Line 423 and 430, please mention the developer name and location for STRUCTURE and DARWIN software.
  • In the statistical analysis, please explain how the data was read (loci/bands produced by each marker) and collected for analysis. For example, “1” for the presence and “0” for absence for data analysis.
  • Please add a separate section of “conclusion”.
  • Are the evaluated germplasm frost-tolerant? If so, please also try to link your discussion with the front tolerant with future directions. In conclusion, also explain how this study can be used for future research.

Reviewer 2 Report

The study described genetic analyzes of 3 Diospyros species, including many commercial varieties and accessions of. D. lotus from different locations that turned out to be very similar to each other and possibly introduced from a single source. The results obtained here could be published somewhere as a short communication, but they do not constitute the basis for an article on this subject due to the insufficient number of polymorphs detected.
Title - I do not think that the phylogenetic relationship of Diospyros germplasm in the Western Caucasus is any different than elsewhere. So this is a mistake.
Abstract - is very vague and indicates that no particularly significant results were obtained in the study.
Introduction - a bit too long. It also points to the lack of a complete understanding of molecular analyzes and identified polymorphism.
The first-time application of the ISSR method in a given species, if  SSR was previously made on it, is not a special achievement. It is obvious that the method will work. Not to mention the fact that repetitive sequences occur in all eukaryotes.
Besides, the authors point to the ISSR815 primer as working on the tested material and the ISSR 813 as not working - it is impossible because they differ only in the anchoring nucleotide. The repeat sequence is the same, so a mistake was made during the analysis in the laboratory.
The method of obtaining the data presented in the additional table no. 2 was not presented in the methodology.
The number of identified polymorphisms is too small to draw far-reaching conclusions. In order to perform the analysis correctly, the number of analyzed loci should be increased, i.e. more primers should be used. The low number of markers used for analysis made the results questionable.

Some additional remarks were included in the file attached.

I can not recommend this article for publishing.
